# Theoretical Investigation of Fast Neutron and Gamma Radiation Properties of Polycarbonate-Bismuth Oxide Composites Using Geant4

**DOI:** 10.3390/nano12203577

**Published:** 2022-10-12

**Authors:** Hanan Akhdar

**Affiliations:** Physics Department, Faculty of Sciences, Imam Mohammad Ibn Saud Islamic University (IMSIU), Riyadh 13318, Saudi Arabia; hfAkdar@imamu.edu.sa

**Keywords:** PC-Bi_2_O_3_, Geant4, gmma, mass attenuation coefficient, half-value layer, shield, fast neutron effective removal cross section

## Abstract

The gamma mass (*µ_m_*) and linear (*µ*) attenuation coefficients of polycarbonate-bismuth oxide composites (PC-Bi_2_O_3_) with different bismuth oxide weight factors were investigated theoretically using EpiXS and a Monte Carlo simulation-based toolkit and Geant4 within an energy range between 0.1 and 2 MeV. The wide energy ranges of gamma rays and neutrons were chosen to cover as many applications as possible. The attenuation coefficients were then used to compute the half-value layers. The effective atomic numbers and effective electron densities of the studied samples obtained by EpiXS were compared as well. In order to further evaluate the shielding effectiveness of the studied samples, the thicknesses of all the investigated samples equivalent to 0.5 mm lead at a gamma energy of 511 keV were compared using a Geant4 code simulating a female numerical phantom with a gamma source placed facing the chest and a cylinder-shaped shield wrapped around the trunk area. The fast neutron removal cross sections of the investigated samples were studied to evaluate the effect of the weight factor of nanocomposites on the neutron shielding capabilities of the polymer as well.

## 1. Introduction

Polymers are flexible, lightweight, and nontoxic materials with low atomic numbers. Adding nano-reinforcements, especially high-density and high-atomic-number metal oxides, to polymers makes them novel gamma shielding materials with many outstanding features [1,2,3]. The photoelectric effect is dominant in high-Z elements, which is why heavy metals such as lead are usually used as a shielding material. However, lead is toxic and heavy, which made researchers try finding better alternatives with comparable shielding properties [4,5].

In this work, fast neutron and gamma attenuation properties of a novel newly introduced polycarbonate-bismuth oxide composite with different weight fractions of nano-Bi_2_O_3_ (0, 5, 10, 20, 30, 40, and 50 wt %) by Mehrara R et al. were theoretically investigated at a wide gamma energy range [1]. EpiXS, which is a Windows-based program for photon attenuation, dosimetry, and shielding, based on the EPICS2017 and EPDL9 database that allows obtaining the photon cross section data for any sample, was used in this work to estimate the attenuation coefficients, half-value layers, effective atomic numbers, and electron densities of all studied samples within the energy range of interest [6]. A Geant4 toolkit, which is based on a Monte Carlo simulation, was also used to investigate the attenuation properties of the samples of interest and compared to those results obtained from EpiXS [7]. Fast neutron removal cross sections were be investigated and compared among the samples as well.

## 2. Materials and Methods

Polycarbonate is a recyclable polymer with an amorphous structure that makes it a great choice when making homogeneous nanocomposites. The densities and composites of the studied samples were as mentioned in a recently published article where the fabrication of different weight percentages of a polycarbonate-bismuth oxide composite was done by Mehrara et al. using a mixed-solution method. The investigated samples are tabulated in Table 1 [1].

## 3. Theory

The mass attenuation coefficient *µ**_m_* is the main parameter investigated when studying the attenuation properties of any sample, and it can be calculated using Equation (1) [8]:(1)$$I=I_{0}e^{-\mu_{m}x}$$
where (*I*_0_) is the mono-energetic incident intensity of photons and (*I*) is the attenuated photons intensity after passing through a mass per unit area (*x*) layer of a certain material. In case the sample is made of mixtures or compounds, Equation (2) can be used [8]:(2)$$\mu_{m}={\sum_{i}w_{i}\left(\mu_{m}\right)}_{i}$$
where (*w_i_*) is the weight of the *i*th element.

The mass attenuation coefficient is very important when it comes to choosing a shielding material. The half-value layer (*HVL*) is also important in predicting the required thickness of a shielding material, and it is sample thickness that reduces the radiation level by a factor of 2, as described by Equation (3).
(3)$$HVL=\frac{\ln2}{\mu }$$
where *μ* (cm^−1^) is the linear attenuation coefficient of the material; the relation between the mass attenuation coefficient and the linear attenuation coefficient is given by Equation (4) [9,10].
(4)$$\mu =\mu_{m}\rho$$

These parameters were studied and compared theoretically for all investigated samples.

Neutron attenuation is described by the neutron-removing cross section (Σ*_R_*) which is the probability of neutron reactions within the material and is given by the mixture rule for each element in the composite material as shown in Equation (5) [11]:(5)$$\sum_{R}={\sum_{i}\rho_{i}\left(\sum_{R}/\rho \right)}_{i}$$
where (*ρ_i_*) is partial density and (Σ*_R_*/*ρ*) is the mass removal cross section, which can be calculated for any compound by Equation (6) [12]:(6)∑Rρ=0.206A−13Z−0.294
where (*A*) is the atomic weight and (*Z*) is the atomic number. The neutron removal coefficient is found by multiplying the neutron removal coefficient by the density of the absorber.

The fast neutron removal cross section of any element can be calculated using the empirical formulas indicated by Equations (7) and (8) [13]:(7)$$\Sigma_{R}=0.190Z^{-0.743}\;\mathrm{if}\;Z\leq 8$$
(8)$$\Sigma_{R}=0.125Z^{-0.565}\;\mathrm{if}\;Z>8$$

## 4. Monte Carlo Simulation

A Geant4 simulation code was developed and used to investigate the desired parameters of the studied samples within the studied energy range. Geant4 is a toolkit based on the Monte Carlo statistical method that simulates the passage of particles in matter [7]. Version 10.07 of Geant4 toolkit was used to develop the code used in the current work. A gamma source placed in front of the sample emitting mono-energetic gamma particles in the direction of the sample followed by a detector surrounded by a lead container. Figure 1 shows a visualization of the simulation code. The ratio between the number of gamma particles that reached the detector with and without the sample was found for each energy; then, the attenuation coefficient based on the sample’s thickness was calculated. This process was repeated for all the studied samples. The results of the Geant4 code were validated using those from the Windows based software; EpiXS.

## 5. Results and Discussion

### 5.1. Attenuation Coefficient

The linear and mass attenuation coefficients were obtained using EpiXS and Geant4 for all studied samples; then, the percentage difference between them was calculated using Equation (9) as shown in Table 2 and Table 3.
(9)$$\%Diff=100\times \left(\mu \mathrm{EpiXS}-\mu G4\right)/\mu \mathrm{EpiXS}$$

Figure 2 and Figure 3 illustrate the mass attenuation coefficients of the studied samples using both EpiXS and Geant4. Root 6.10/04 software was used to plot them [14].

The results showed that increasing the weight factor of the nanocomposites in the polymer increased the ability to shield against gamma rays, which is very clear from Figure 2 and Figure 3, especially at lower energies. The mass attenuation coefficients of the studied samples were used to compute the half-value layers of the samples within the studied energy range.

### 5.2. Half-Value Layer (HVL)

The half-value layer is an important parameter for any radiation shielding design since it refers to the required thickness of an absorber to reduce the radiation level to half of its initial value. Table 3 summarizes all the HVLs of the studied samples.

The results showed that increasing the weight factor of the nanocomposites in the polymer decreased the half-value layer, which is very notable in Figure 3. The shielding properties of the polycarbonate-bismuth oxide composite were very promising, especially when adding the composites’ weight fractions. The mass attenuation coefficient values increased as the weight fractions of the Bi_2_O_3_ nanocomposites increased noticeably. Geant4 results showed very good agreement with those obtained from EpiXS. Further studies of different composites at different energy ranges may be necessary to fully cover the shielding capabilities of polycarbonate-bismuth oxide.

### 5.3. Effective Atomic Number (Z_eff_) and Effective Electron Density (N_eff_)

The effective atomic number and effective electron density of all investigated samples were found using EpiXS as shown in Table 4 and Table 5 and plotted in Figure 4 and Figure 5.

The Z_eff_ and N_eff_ of the investigated samples were evaluated using EpiXS. The results showed that they both behaved in a similar way and that Z_eff_ and N_eff_ increased with increments of nanocomposites’ weight factors along the whole energy range studied. These results agree with previously published results that nanocomposites have an impressive shielding effect from low-energy gamma radiation, and here we conclude that this applies on a wide range of gamma radiation as well.

## 6. Fast Neutron Removal Cross Section

The fast neutron removal cross sections were found using a Geant4 code and then calculated for all the studied samples using Equations (5), (7) and (8) with the use of the fast neutron removal cross sections and the weight fractions of each element in each polymer [15,16,17]. Table 6 lists the Geant4-obtained fast neutron removal cross sections compared with those calculated in order to validate the obtained results.

Good agreement was shown between the calculated removal cross sections and those found using Geant4 as the percentage deviation was less than 9% and the agreement became better as the weight factor of the nanocomposites increased. The fast neutron removal cross section increased as the weight fraction of the nanocomposites increased in the polymer, as can be seen from the obtained results. This concludes that the weight factor of the nanocomposites increased the neutron shielding ability of the polymer.

## 7. Lead-Equivalent Gamma Shield

The shielding thicknesses equivalent to 0.5 mm of lead of all studied samples were simulated using the Geant4 toolkit with a numerical female human phantom and a 511 keV gamma source placed in front of it. Table 7 summarizes the thicknesses of the studied samples equivalent to 0.5 mm of lead with a gamma energy of 511 keV.

Figure 6 represents the simulation visualization of the phantom with the gamma source and the simulated shield covering the whole trunk area. Table 8 shows the energy deposit with and without the shields in the whole body, the head, and the trunk. Percentage of energy loss with the shield compared to that without are shown in Figure 7.

The energy deposit in the head of the phantom which was not shielded as well as the total energy deposit in the trunk of the phantom were both compared without and with 0.5 mm-lead-equivalent cylindrical shielding. The obtained results showed that the energy delivered to the head increased while using the shields, which may be due to scattering of the gamma rays when hitting the material of the shields. Meanwhile, the total energy delivered to the trunk was reduced when using the shields, although the detailed energy deposit in the organs may have varied. Table 8 show the energy deposit in the head, trunk, and whole body with and without the studied shields.

The results of the simulation code showed that using PC-Bi_2_O_3_ as a shield decreased the energy deposit within the protected areas even more than lead shielding did. The energy deposit of the trunk area which was fully covered by the shield was less than the energy deposit with no shield by 6 to 8%, whereas it was decreased by only 0.88% in case of a lead shield. The energy deposit of the head, which was not protected at all, increased in all cases by 14 to 29% which was due to the scattered gamma rays. The energy deposit in the whole body was decreased by 6 to 7% when using PC-Bi_2_O_3_ as a shield, while it decreased only by 0.29% when using a traditional lead shield.

## 8. Conclusions

The results showed that the gamma shielding properties of the studied samples became better with an increasing weight factor of the Bi_2_O_3_ nanocomposites within the PC, especially at lower gamma energies, and it became almost equal at high energies. This means that PC-Bi_2_O_3_ could be used as a replacement to toxic traditional lead shielding materials. The results of the second simulation code showed that using PC-Bi_2_O_3_ as a shield decreased the energy deposit within the protected areas as well as the whole body more than a traditional lead shield did. The fast neutron shielding capabilities of PC-Bi_2_O_3_ were increased as well when the weight factor of the Bi_2_O_3_ nanocomposites was increased.

The obtained results show that PC-Bi_2_O_3_ is a good candidate to replace traditional gamma and neutron shielding material, and further studies at other gamma and neutron energies may agree with these results. The results also show that Geant4 could be used in estimating the shielding properties of materials.

## Figures and Tables

**Figure 1 nanomaterials-12-03577-f001:**
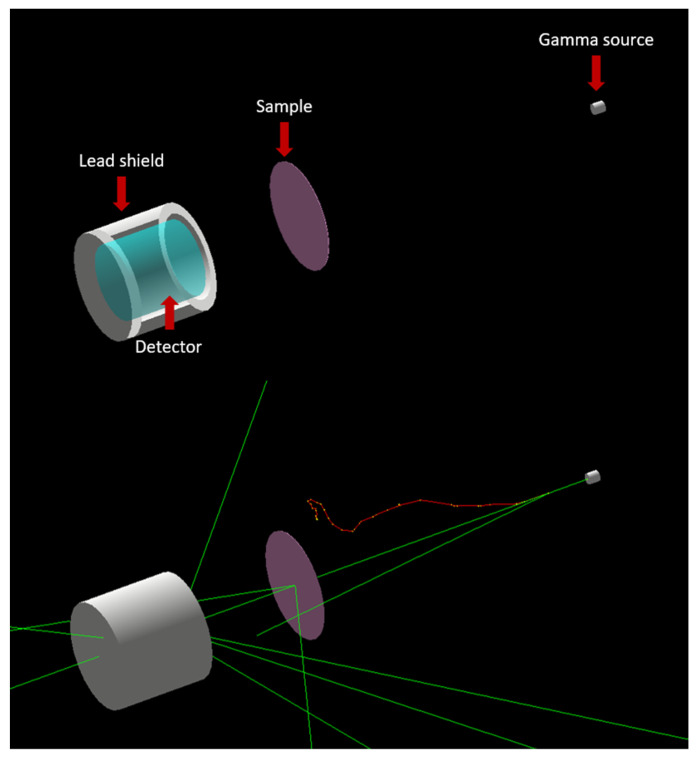
Simulation code visualization (green lines indicate the paths of non-charged particles and red lines indicate the paths of negatively charged particles).

**Figure 2 nanomaterials-12-03577-f002:**
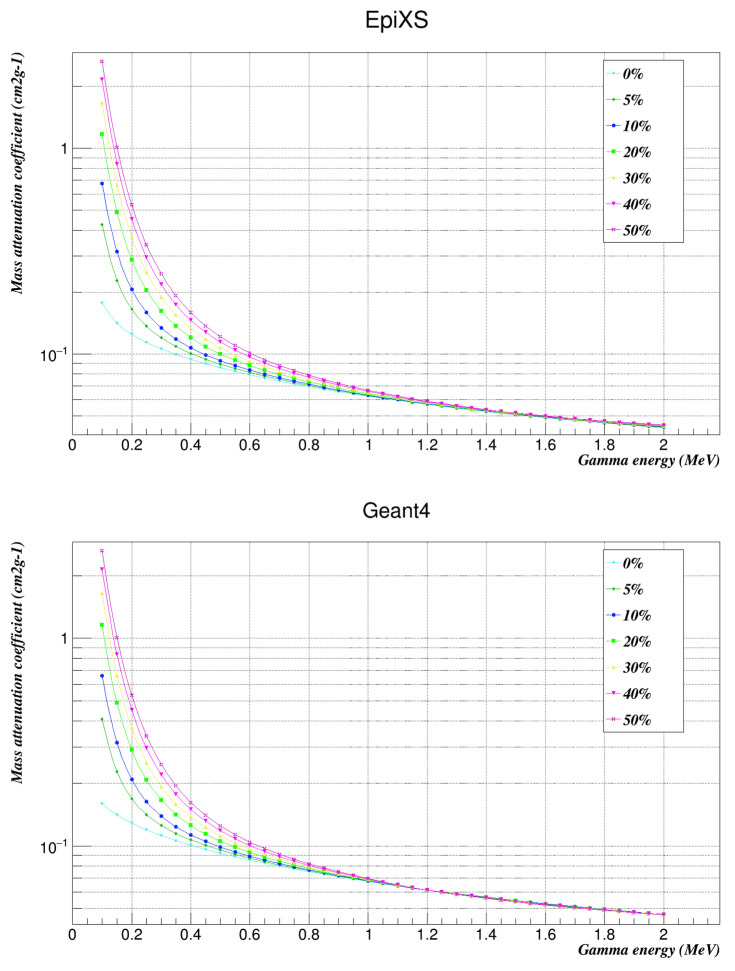
Mass attenuation coefficients of the investigated samples at the studied energies as found by EpiXS and Geant4.

**Figure 3 nanomaterials-12-03577-f003:**
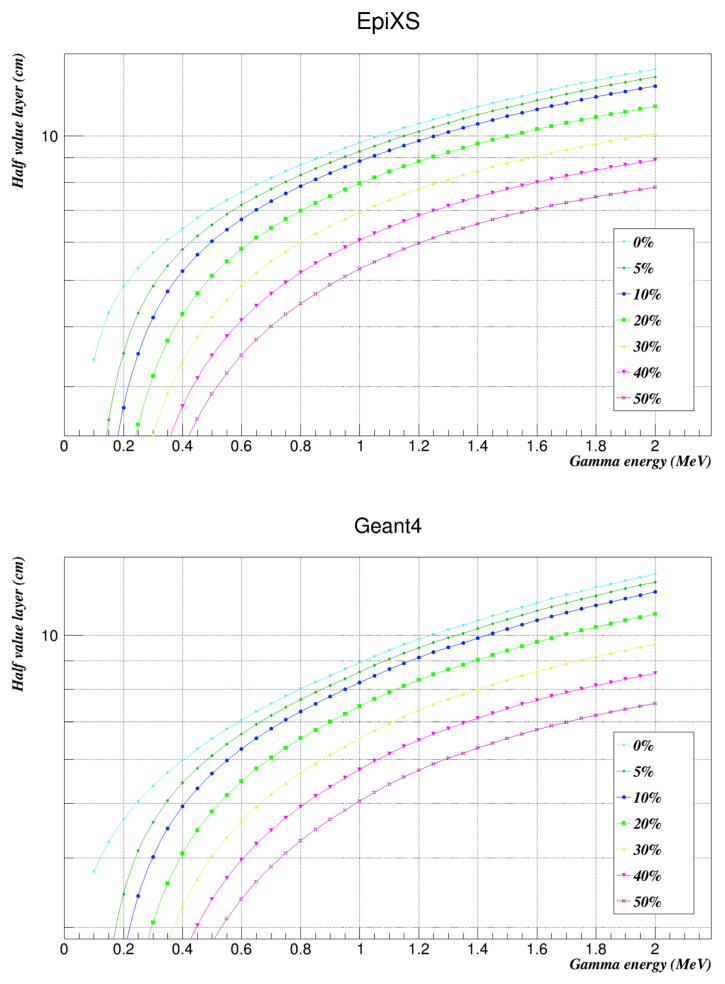
Half-value layers of the investigated samples at the studied energies as found by EpiXS and Geant4.

**Figure 4 nanomaterials-12-03577-f004:**
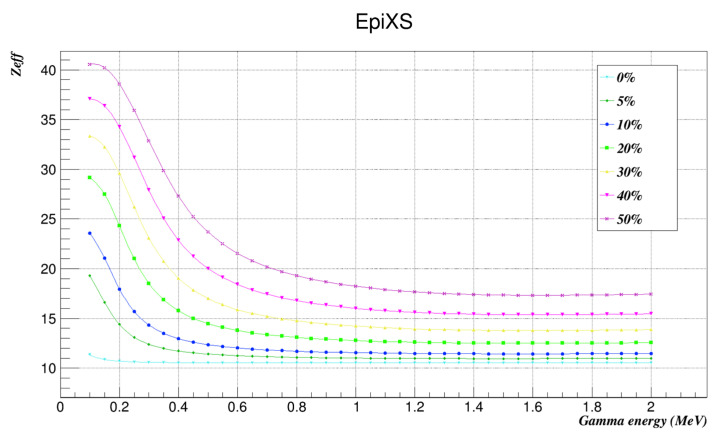
The effective atomic numbers of the investigated samples at the studied energies as found by EpiXS.

**Figure 5 nanomaterials-12-03577-f005:**
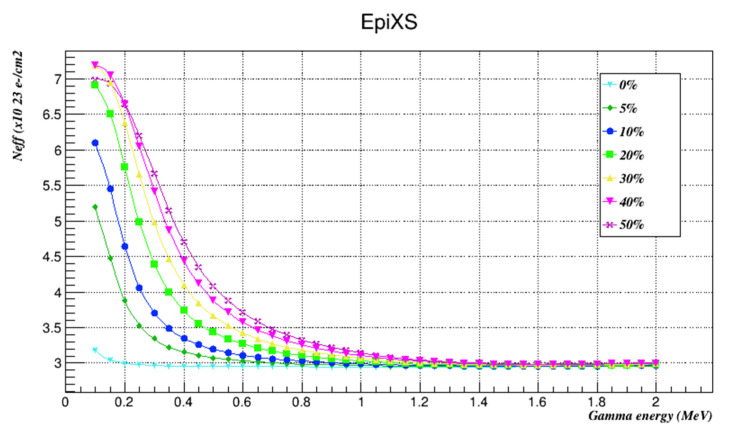
The effective electron densities of the investigated samples at the studied energies as found by EpiXS.

**Figure 6 nanomaterials-12-03577-f006:**
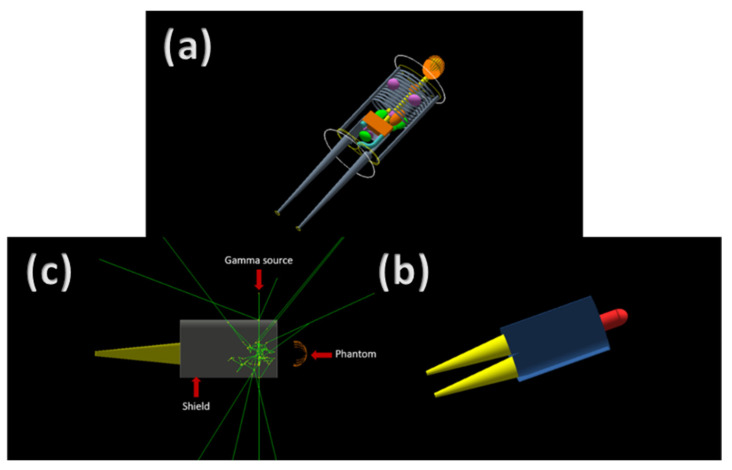
Geant4 visualization of the simulated phantom with a shield (**a**) wire frame; (**b**) solid view, (**c**) side view with the gamma source shooting gamma particles in the direction of the phantom.

**Figure 7 nanomaterials-12-03577-f007:**
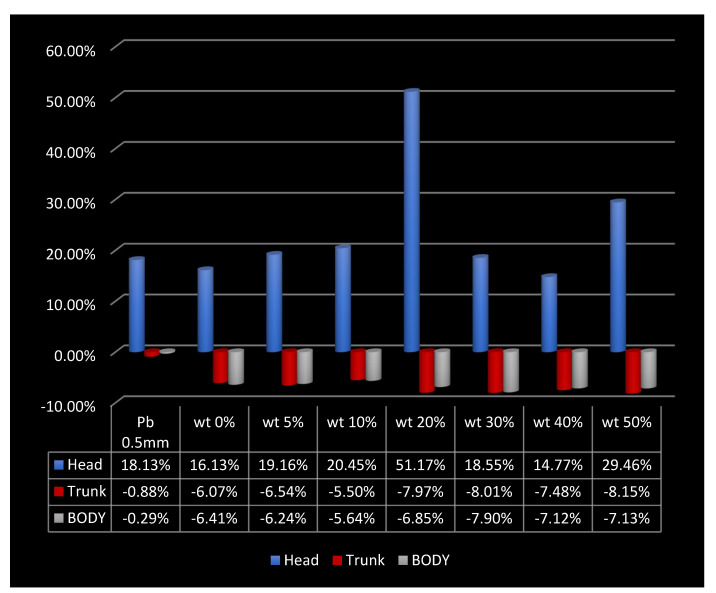
Percentages of energy deposit to the body, head, and trunk with different shields.

**Table 1 nanomaterials-12-03577-t001:** A list of the studied samples’ densities.

PC wt %	Bi_2_O_3_ wt %	Density (gm/cm^3^)
100	0	1.15
95	5	1.19
90	10	1.24
80	20	1.36
70	30	1.54
60	40	1.74
50	50	1.97

**Table 2 nanomaterials-12-03577-t002:** Mass attenuation coefficients of the samples found by EpiXS and Geant4.

	PC-Bi_2_O_3_ wt%											
Energy (MeV)	0%				5%				10%			
	**Mass attenuation coefficients (cm^2^ g^−1^)**											
	**EpiXS**		**Geant4**	**Diff%**	**EpiXS**	**Geant4**	**Diff%**	**EpiXS**		**Geant4**		**Diff%**
**0.10**	0.17684		0.15972	9.68%	0.42520	0.40872	3.88%	0.67357		0.65773		2.35%
**0.15**	0.14085		0.14121	−0.26%	0.22783	0.22784	−0.01%	0.31480		0.31447		0.10%
**0.20**	0.12444		0.12881	−3.51%	0.16526	0.16916	−2.36%	0.20607		0.20950		−1.67%
**0.25**	0.11373		0.11953	−5.10%	0.13632	0.14157	−3.85%	0.15891		0.16360		−2.95%
**0.30**	0.10576		0.11215	−6.03%	0.11969	0.12570	−5.02%	0.13362		0.13925		−4.21%
**0.35**	0.09932		0.10603	−6.75%	0.10858	0.11491	−5.83%	0.11785		0.12380		−5.05%
**0.40**	0.09406		0.10082	−7.18%	0.10055	0.10694	−6.36%	0.10705		0.11307		−5.63%
**0.45**	0.08949		0.09628	−7.58%	0.09424	0.10070	−6.86%	0.09898		0.10512		−6.20%
**0.50**	0.08561		0.09228	−7.79%	0.08920	0.09554	−7.11%	0.09278		0.09879		−6.47%
**0.55**	0.08213		0.08871	−8.01%	0.08491	0.09116	−7.36%	0.08769		0.09361		−6.75%
**0.60**	0.07906		0.08547	−8.11%	0.08124	0.08735	−7.52%	0.08343		0.08923		−6.95%
**0.65**	0.07626		0.08254	−8.23%	0.07800	0.08399	−7.68%	0.07974		0.08544		−7.16%
**0.70**	0.07374		0.07988	−8.34%	0.07514	0.08101	−7.81%	0.07654		0.08214		−7.31%
**0.75**	0.07143		0.07745	−8.43%	0.07258	0.07833	−7.93%	0.07372		0.07921		−7.45%
**0.80**	0.06931		0.07522	−8.52%	0.07025	0.07590	−8.04%	0.07119		0.07658		−7.57%
**0.85**	0.06733		0.07314	−8.64%	0.06810	0.07367	−8.17%	0.06887		0.07419		−7.71%
**0.90**	0.06551		0.07119	−8.67%	0.06615	0.07159	−8.21%	0.06680		0.07198		−7.77%
**0.95**	0.06384		0.06933	−8.59%	0.06437	0.06962	−8.15%	0.06490		0.06991		−7.72%
**1.00**	0.06225		0.06752	−8.46%	0.06269	0.06773	−8.03%	0.06313		0.06794		−7.61%
**1.05**	0.06074		0.06574	−8.22%	0.06111	0.06588	−7.81%	0.06147		0.06602		−7.40%
**1.10**	0.05936		0.06412	−8.01%	0.05966	0.06421	−7.62%	0.05996		0.06429		−7.22%
**1.15**	0.05802		0.06264	−7.97%	0.05827	0.06268	−7.58%	0.05851		0.06272		−7.19%
**1.20**	0.05683		0.06129	−7.85%	0.05703	0.06129	−7.47%	0.05724		0.06129		−7.09%
**1.25**	0.05565		0.06004	−7.88%	0.05583	0.06001	−7.50%	0.05600		0.05999		−7.12%
**1.30**	0.05454		0.05889	−7.97%	0.05469	0.05884	−7.58%	0.05483		0.05879		−7.21%
**1.35**	0.05349		0.05782	−8.09%	0.05361	0.05775	−7.71%	0.05374		0.05768		−7.33%
**1.40**	0.05252		0.05678	−8.13%	0.05263	0.05670	−7.75%	0.05274		0.05662		−7.37%
**1.45**	0.05155		0.05568	−8.01%	0.05165	0.05559	−7.64%	0.05175		0.05551		−7.26%
**1.50**	0.05070		0.05464	−7.78%	0.05079	0.05455	−7.41%	0.05088		0.05447		−7.04%
**1.55**	0.05058		0.05367	−6.11%	0.04995	0.05358	−7.27%	0.05004		0.05349		−6.90%
**1.60**	0.04974		0.05275	−6.04%	0.04913	0.05266	−7.20%	0.04922		0.05258		−6.83%
**1.65**	0.04897		0.05188	−5.94%	0.04835	0.05180	−7.14%	0.04844		0.05172		−6.78%
**1.70**	0.04825		0.05105	−5.82%	0.04761	0.05098	−7.08%	0.04770		0.05090		−6.71%
**1.75**	0.04758		0.05027	−5.64%	0.04692	0.05020	−6.98%	0.04702		0.05013		−6.62%
**1.80**	0.04694		0.04952	−5.50%	0.04624	0.04946	−6.95%	0.04634		0.04940		−6.59%
**1.85**	0.04633		0.04881	−5.34%	0.04560	0.04875	−6.91%	0.04571		0.04870		−6.55%
**1.90**	0.04579		0.04812	−5.08%	0.04502	0.04808	−6.79%	0.04513		0.04803		−6.43%
**1.95**	0.04526		0.04745	−4.85%	0.04444	0.04742	−6.71%	0.04456		0.04738		−6.35%
**2.00**	0.04474		0.04680	−4.59%	0.04386	0.04677	−6.63%	0.04399		0.04675		−6.28%
Energy (MeV)	20%			30%			40%			50%		
	**Mass attenuation coefficients (cm^2^ g^−1^)**											
	**EpiXS**	**Geant4**	**Diff%**	**EpiXS**	**Geant4**	**Diff%**	**EpiXS**	**Geant4**	**Diff%**	**EpiXS**	**Geant4**	**Diff%**
**0.10**	1.17030	1.15574	1.24%	1.66703	1.65375	0.80%	2.16376	2.15176	0.55%	2.66049	2.64977	0.40%
**0.15**	0.48876	0.48774	0.21%	0.66271	0.66100	0.26%	0.83666	0.83426	0.29%	1.01061	1.00752	0.31%
**0.20**	0.28769	0.29020	−0.87%	0.36932	0.37089	−0.43%	0.45094	0.45158	−0.14%	0.53257	0.53228	0.05%
**0.25**	0.20408	0.20767	−1.76%	0.24925	0.25173	−1.00%	0.29443	0.29580	−0.47%	0.33960	0.33987	−0.08%
**0.30**	0.16148	0.16636	−3.02%	0.18934	0.19347	−2.18%	0.21720	0.22058	−1.56%	0.24506	0.24769	−1.08%
**0.35**	0.13637	0.14157	−3.81%	0.15489	0.15934	−2.87%	0.17342	0.17712	−2.13%	0.19194	0.19489	−1.54%
**0.40**	0.12003	0.12533	−4.41%	0.13302	0.13759	−3.44%	0.14601	0.14985	−2.63%	0.15900	0.16211	−1.96%
**0.45**	0.10847	0.11396	−5.06%	0.11796	0.12280	−4.10%	0.12745	0.13164	−3.29%	0.13694	0.14048	−2.59%
**0.50**	0.09995	0.10529	−5.34%	0.10712	0.11179	−4.37%	0.11428	0.11830	−3.51%	0.12145	0.12480	−2.76%
**0.55**	0.09325	0.09851	−5.64%	0.09881	0.10341	−4.66%	0.10437	0.10832	−3.78%	0.10993	0.11322	−2.99%
**0.60**	0.08779	0.09298	−5.91%	0.09216	0.09673	−4.96%	0.09653	0.10048	−4.10%	0.10089	0.10424	−3.32%
**0.65**	0.08321	0.08835	−6.17%	0.08668	0.09125	−5.27%	0.09016	0.09415	−4.43%	0.09363	0.09705	−3.66%
**0.70**	0.07935	0.08439	−6.36%	0.08216	0.08665	−5.47%	0.08496	0.08890	−4.64%	0.08777	0.09116	−3.86%
**0.75**	0.07600	0.08096	−6.53%	0.07828	0.08272	−5.66%	0.08057	0.08447	−4.85%	0.08285	0.08623	−4.08%
**0.80**	0.07307	0.07794	−6.67%	0.07494	0.07930	−5.81%	0.07682	0.08066	−5.00%	0.07869	0.08202	−4.22%
**0.85**	0.07042	0.07523	−6.83%	0.07197	0.07628	−5.99%	0.07351	0.07732	−5.18%	0.07506	0.07837	−4.40%
**0.90**	0.06808	0.07277	−6.90%	0.06936	0.07356	−6.06%	0.07064	0.07435	−5.26%	0.07192	0.07515	−4.48%
**0.95**	0.06596	0.07050	−6.88%	0.06702	0.07108	−6.07%	0.06808	0.07167	−5.28%	0.06914	0.07226	−4.51%
**1.00**	0.06401	0.06836	−6.79%	0.06489	0.06878	−5.99%	0.06577	0.06919	−5.21%	0.06665	0.06961	−4.45%
**1.05**	0.06220	0.06631	−6.60%	0.06293	0.06659	−5.82%	0.06366	0.06688	−5.06%	0.06439	0.06716	−4.31%
**1.10**	0.06056	0.06447	−6.45%	0.06116	0.06464	−5.69%	0.06176	0.06481	−4.95%	0.06236	0.06499	−4.22%
**1.15**	0.05901	0.06280	−6.42%	0.05951	0.06288	−5.66%	0.06000	0.06295	−4.92%	0.06050	0.06303	−4.19%
**1.20**	0.05764	0.06130	−6.34%	0.05805	0.06130	−5.59%	0.05846	0.06130	−4.86%	0.05887	0.06131	−4.14%
**1.25**	0.05634	0.05993	−6.37%	0.05668	0.05988	−5.63%	0.05703	0.05982	−4.90%	0.05737	0.05976	−4.18%
**1.30**	0.05513	0.05868	−6.46%	0.05542	0.05858	−5.71%	0.05571	0.05848	−4.98%	0.05600	0.05838	−4.25%
**1.35**	0.05399	0.05754	−6.58%	0.05425	0.05741	−5.83%	0.05450	0.05727	−5.09%	0.05475	0.05713	−4.35%
**1.40**	0.05296	0.05646	−6.61%	0.05319	0.05630	−5.87%	0.05341	0.05615	−5.12%	0.05363	0.05599	−4.39%
**1.45**	0.05195	0.05534	−6.52%	0.05215	0.05517	−5.78%	0.05235	0.05500	−5.04%	0.05256	0.05483	−4.32%
**1.50**	0.05107	0.05429	−6.30%	0.05126	0.05412	−5.57%	0.05145	0.05394	−4.84%	0.05164	0.05377	−4.12%
**1.55**	0.05022	0.05332	−6.17%	0.05040	0.05314	−5.45%	0.05058	0.05297	−4.73%	0.05076	0.05280	−4.01%
**1.60**	0.04939	0.05241	−6.11%	0.04957	0.05224	−5.39%	0.04974	0.05207	−4.67%	0.04992	0.05190	−3.97%
**1.65**	0.04861	0.05156	−6.05%	0.04879	0.05140	−5.34%	0.04897	0.05123	−4.63%	0.04915	0.05107	−3.92%
**1.70**	0.04788	0.05075	−6.00%	0.04806	0.05060	−5.28%	0.04825	0.05045	−4.57%	0.04843	0.05030	−3.87%
**1.75**	0.04721	0.05000	−5.91%	0.04740	0.04986	−5.20%	0.04758	0.04972	−4.50%	0.04777	0.04959	−3.80%
**1.80**	0.04654	0.04928	−5.88%	0.04674	0.04916	−5.17%	0.04694	0.04903	−4.47%	0.04714	0.04891	−3.77%
**1.85**	0.04591	0.04860	−5.84%	0.04612	0.04849	−5.13%	0.04633	0.04839	−4.43%	0.04654	0.04828	−3.74%
**1.90**	0.04535	0.04795	−5.72%	0.04557	0.04786	−5.02%	0.04579	0.04777	−4.32%	0.04601	0.04768	−3.63%
**1.95**	0.04479	0.04732	−5.64%	0.04502	0.04725	−4.94%	0.04526	0.04718	−4.25%	0.04549	0.04711	−3.56%
**2.00**	0.04424	0.04670	−5.57%	0.04449	0.04666	−4.87%	0.04474	0.04661	−4.18%	0.04499	0.04656	−3.49%

**Table 3 nanomaterials-12-03577-t003:** Half-value layers of the samples found by EpiXS and Geant4.

	PC-Bi_2_O_3_ wt%											
Energy (MeV)	0%				5%				10%			
	**HVL (cm)**											
	**EpiXS**		**Geant4**	**Diff%**	**EpiXS**	**Geant4**	**Diff%**		**EpiXS**	**Geant4**		**Diff%**
**0.10**	3.40842		3.77378	−10.72%	1.36988	1.42512	−4.03%		0.82989	0.84988		−2.41%
**0.15**	4.27929		4.26831	0.26%	2.55667	2.55648	0.01%		1.77568	1.77754		−0.10%
**0.20**	4.84351		4.67931	3.39%	3.52472	3.44345	2.31%		2.71265	2.66818		1.64%
**0.25**	5.29959		5.04247	4.85%	4.27288	4.11455	3.71%		3.51773	3.41683		2.87%
**0.30**	5.69886		5.37462	5.69%	4.86640	4.63386	4.78%		4.18335	4.01417		4.04%
**0.35**	6.06855		5.68480	6.32%	5.36432	5.06889	5.51%		4.74341	4.51534		4.81%
**0.40**	6.40811		5.97864	6.70%	5.79278	5.44656	5.98%		5.22196	4.94362		5.33%
**0.45**	6.73512		6.26031	7.05%	6.18101	5.78433	6.42%		5.64742	5.31768		5.84%
**0.50**	7.04013		6.53130	7.23%	6.53017	6.09695	6.63%		6.02482	5.65853		6.08%
**0.55**	7.33883		6.79474	7.41%	6.85996	6.38978	6.85%		6.37464	5.97156		6.32%
**0.60**	7.62396		7.05187	7.50%	7.16968	6.66843	6.99%		6.70049	6.26494		6.50%
**0.65**	7.90326		7.30213	7.61%	7.46756	6.93477	7.13%		7.01037	6.54212		6.68%
**0.70**	8.17428		7.54537	7.69%	7.75198	7.19023	7.25%		7.30301	6.80554		6.81%
**0.75**	8.43770		7.78204	7.77%	8.02584	7.43623	7.35%		7.58296	7.05734		6.93%
**0.80**	8.69585		8.01318	7.85%	8.29134	7.67446	7.44%		7.85217	7.29962		7.04%
**0.85**	8.95225		8.24047	7.95%	8.55311	7.90703	7.55%		8.11608	7.53478		7.16%
**0.90**	9.20013		8.46633	7.98%	8.80478	8.13657	7.59%		8.36869	7.76560		7.21%
**0.95**	9.44089		8.69392	7.91%	9.04854	8.36637	7.54%		8.61286	7.99541		7.17%
**1.00**	9.68233		8.92725	7.80%	9.29128	8.60045	7.44%		8.85456	8.22817		7.07%
**1.05**	9.92275		9.16877	7.60%	9.53205	8.84141	7.25%		9.09348	8.46660		6.89%
**1.10**	10.15320		9.39993	7.42%	9.76271	9.07173	7.08%		9.32232	8.69421		6.74%
**1.15**	10.38880		9.62170	7.38%	9.99685	9.29254	7.05%		9.55311	8.91232		6.71%
**1.20**	10.60670		9.83422	7.28%	10.21340	9.50338	6.95%		9.76655	9.11991		6.62%
**1.25**	10.82990		10.03848	7.31%	10.43380	9.70555	6.98%		9.98243	9.31853		6.65%
**1.30**	11.05050		10.23518	7.38%	10.65060	9.89977	7.05%		10.19410	9.50887		6.72%
**1.35**	11.26860		10.42474	7.49%	10.86420	10.08626	7.16%		10.40160	9.69104		6.83%
**1.40**	11.47740		10.61483	7.52%	11.06810	10.27243	7.19%		10.59930	9.87208		6.86%
**1.45**	11.69300		10.82550	7.42%	11.27790	10.47765	7.10%		10.80200	10.07062		6.77%
**1.50**	11.88930		11.03077	7.22%	11.46830	10.67708	6.90%		10.98550	10.26302		6.58%
**1.55**	12.08910		11.23105	7.10%	11.66170	10.87120	6.78%		11.17130	10.44985		6.46%
**1.60**	12.29100		11.42679	7.03%	11.85650	11.06045	6.71%		11.35800	10.63157		6.40%
**1.65**	12.48990		11.61833	6.98%	12.04790	11.24524	6.66%		11.54090	10.80860		6.35%
**1.70**	12.68430		11.80603	6.92%	12.23450	11.42595	6.61%		11.71870	10.98138		6.29%
**1.75**	12.87060		11.99020	6.84%	12.41310	11.60296	6.53%		11.88870	11.15029		6.21%
**1.80**	13.06180		12.17131	6.82%	12.59570	11.77665	6.50%		12.06200	11.31572		6.19%
**1.85**	13.24780		12.34969	6.78%	12.77320	11.94743	6.46%		12.23020	11.47806		6.15%
**1.90**	13.42130		12.52575	6.67%	12.93830	12.11567	6.36%		12.38620	11.63767		6.04%
**1.95**	13.59950		12.70198	6.60%	13.10770	12.28374	6.29%		12.54600	11.79681		5.97%
**2.00**	13.78030		12.87984	6.53%	13.27900	12.45308	6.22%		12.70720	11.95690		5.90%
Energy (MeV)	20%			30%			40%			50%		
	**HVL (cm)**											
	**EpiXS**	**Geant4**	**Diff%**	**EpiXS**	**Geant4**	**Diff%**	**EpiXS**	**Geant4**	**Diff%**	**EpiXS**	**Geant4**	**Diff%**
**0.10**	0.43550	0.44099	−1.26%	0.27000	0.27217	−0.80%	0.18411	0.18513	−0.56%	0.13225	0.13279	−0.40%
**0.15**	1.04279	1.04496	−0.21%	0.67918	0.68093	−0.26%	0.47613	0.47750	−0.29%	0.34816	0.34923	−0.31%
**0.20**	1.77157	1.75629	0.86%	1.21872	1.21356	0.42%	0.88339	0.88214	0.14%	0.66067	0.66103	−0.05%
**0.25**	2.49739	2.45425	1.73%	1.80578	1.78798	0.99%	1.35300	1.34672	0.46%	1.03608	1.03526	0.08%
**0.30**	3.15621	3.06358	2.93%	2.37720	2.32640	2.14%	1.83410	1.80595	1.53%	1.43581	1.42053	1.06%
**0.35**	3.73739	3.60008	3.67%	2.90583	2.82468	2.79%	2.29711	2.24914	2.09%	1.83311	1.80538	1.51%
**0.40**	4.24603	4.06654	4.23%	3.38364	3.27126	3.32%	2.72833	2.65839	2.56%	2.21295	2.17046	1.92%
**0.45**	4.69864	4.47237	4.82%	3.81563	3.66527	3.94%	3.12560	3.02614	3.18%	2.56937	2.50464	2.52%
**0.50**	5.09933	4.84060	5.07%	4.20200	4.02619	4.18%	3.48579	3.36752	3.39%	2.89714	2.81939	2.68%
**0.55**	5.46563	5.17372	5.34%	4.55520	4.35241	4.45%	3.81685	3.67780	3.64%	3.20072	3.10776	2.90%
**0.60**	5.80539	5.48158	5.58%	4.88390	4.65305	4.73%	4.12698	3.96442	3.94%	3.48738	3.37546	3.21%
**0.65**	6.12502	5.76894	5.81%	5.19238	4.93262	5.00%	4.41851	4.23108	4.24%	3.75788	3.62535	3.53%
**0.70**	6.42312	6.03920	5.98%	5.47859	5.19447	5.19%	4.68869	4.48076	4.43%	4.00886	3.85969	3.72%
**0.75**	6.70620	6.29517	6.13%	5.74965	5.44144	5.36%	4.94458	4.71595	4.62%	4.24695	4.08059	3.92%
**0.80**	6.97550	6.53942	6.25%	6.00597	5.67606	5.49%	5.18581	4.93895	4.76%	4.47116	4.29001	4.05%
**0.85**	7.23746	6.77458	6.40%	6.25419	5.90083	5.65%	5.41888	5.15203	4.92%	4.68762	4.48988	4.22%
**0.90**	7.48665	7.00348	6.45%	6.48944	6.11843	5.72%	5.63933	5.35758	5.00%	4.89219	4.68229	4.29%
**0.95**	7.72688	7.22940	6.44%	6.71596	6.33184	5.72%	5.85159	5.55827	5.01%	5.08928	4.86956	4.32%
**1.00**	7.96241	7.45611	6.36%	6.93649	6.54446	5.65%	6.05715	5.75710	4.95%	5.27941	5.05431	4.26%
**1.05**	8.19399	7.68638	6.19%	7.15247	6.75893	5.50%	6.25789	5.95656	4.82%	5.46473	5.23881	4.13%
**1.10**	8.41581	7.90580	6.06%	7.35945	6.96305	5.39%	6.45045	6.14625	4.72%	5.64269	5.41421	4.05%
**1.15**	8.63702	8.11590	6.03%	7.56395	7.15845	5.36%	6.63922	6.32783	4.69%	5.81602	5.58218	4.02%
**1.20**	8.84162	8.31474	5.96%	7.75318	7.34247	5.30%	6.81401	6.49814	4.64%	5.97665	5.73914	3.97%
**1.25**	9.04629	8.50422	5.99%	7.94065	7.51722	5.33%	6.98572	6.65936	4.67%	6.13329	5.88737	4.01%
**1.30**	9.24564	8.68501	6.06%	8.12217	7.68333	5.40%	7.15110	6.81212	4.74%	6.28343	6.02737	4.08%
**1.35**	9.43944	8.85694	6.17%	8.29731	7.84035	5.51%	7.30956	6.95572	4.84%	6.42638	6.15833	4.17%
**1.40**	9.62328	9.02641	6.20%	8.46280	7.99391	5.54%	7.45874	7.09513	4.87%	6.56049	6.28459	4.21%
**1.45**	9.81059	9.21031	6.12%	8.63035	8.15892	5.46%	7.60888	7.24348	4.80%	6.69471	6.41769	4.14%
**1.50**	9.97924	9.38764	5.93%	8.78042	8.31719	5.28%	7.74270	7.38509	4.62%	6.81377	6.54410	3.96%
**1.55**	10.14910	9.55895	5.81%	8.93087	8.46936	5.17%	7.87621	7.52053	4.52%	6.93199	6.66444	3.86%
**1.60**	10.31880	9.72480	5.76%	9.08031	8.61598	5.11%	8.00809	7.65045	4.47%	7.04814	6.77929	3.81%
**1.65**	10.48420	9.88568	5.71%	9.22510	8.75754	5.07%	8.13516	7.77528	4.42%	7.15943	6.88916	3.78%
**1.70**	10.64410	10.04201	5.66%	9.36437	8.89454	5.02%	8.25672	7.89556	4.37%	7.26532	6.99453	3.73%
**1.75**	10.79650	10.19424	5.58%	9.49669	9.02739	4.94%	8.37188	8.01174	4.30%	7.36532	7.09587	3.66%
**1.80**	10.95090	10.34277	5.55%	9.62994	9.15649	4.92%	8.48708	8.12413	4.28%	7.46471	7.19350	3.63%
**1.85**	11.10040	10.48793	5.52%	9.75861	9.28214	4.88%	8.59804	8.23307	4.24%	7.56017	7.28769	3.60%
**1.90**	11.23820	10.63009	5.41%	9.87654	9.40468	4.78%	8.69911	8.33884	4.14%	7.64657	7.37872	3.50%
**1.95**	11.37910	10.77126	5.34%	9.99668	9.52587	4.71%	8.80176	8.44299	4.08%	7.73404	7.46794	3.44%
**2.00**	11.52030	10.91273	5.27%	10.11640	9.64680	4.64%	8.90339	8.54649	4.01%	7.82005	7.55621	3.37%

**Table 4 nanomaterials-12-03577-t004:** The effective atomic number of the investigated samples at the studied gamma energy range.

Energy (MeV)	Z_eff_						
	**PC-Bi_2_O_3_ wt%**						
	**0%**	**5%**	**10%**	**20%**	**30%**	**40%**	**50%**
0.1	11.35382	19.29335	23.55752	29.15661	33.35804	37.05912	40.55943
0.15	10.87042	16.63430	21.07820	27.46215	32.24854	36.37394	40.21914
0.2	10.70026	14.39441	17.95276	24.29631	29.59574	34.24601	38.55636
0.25	10.62638	13.10791	15.69913	20.99781	26.21363	31.17795	35.94044
0.3	10.58977	12.40470	14.33301	18.52490	23.10089	27.90935	32.86597
0.35	10.56876	11.97727	13.50051	16.87030	20.72968	25.07236	29.86513
0.4	10.55574	11.71610	12.95348	15.76420	19.03224	22.86361	27.28675
0.45	10.54807	11.54008	12.60329	15.00345	17.84653	21.22975	25.25100
0.5	10.54304	11.41588	12.35222	14.47573	16.99372	20.01411	23.69182
0.55	10.53723	11.32242	12.16448	14.08286	16.37336	19.12569	22.50555
0.6	10.53438	11.25084	12.01780	13.78165	15.88193	18.41867	21.54421
0.65	10.53119	11.19369	11.90808	13.54577	15.49857	17.85468	20.78446
0.7	10.52890	11.15008	11.82536	13.36086	15.19569	17.42795	20.18178
0.75	10.52695	11.11416	11.75679	13.20689	14.94703	17.06603	19.69079
0.8	10.52552	11.08539	11.70145	13.08272	14.75495	16.78125	19.29234
0.85	10.52439	11.06131	11.65513	12.97922	14.59154	16.54098	18.95319
0.9	10.52345	11.04123	11.61642	12.89797	14.45406	16.33817	18.67776
0.95	10.52263	11.02365	11.58229	12.82640	14.33264	16.15801	18.43245
1	10.52200	11.00913	11.55429	12.76713	14.23135	16.00824	18.22736
1.05	10.52138	10.99676	11.52961	12.71507	14.14344	15.88412	18.04725
1.1	10.52091	10.98628	11.50758	12.66838	14.06336	15.77211	17.89013
1.15	10.52060	10.97797	11.48977	12.63043	13.99911	15.68124	17.76436
1.2	10.52050	10.97078	11.47437	12.59738	13.94562	15.60087	17.65304
1.25	10.52062	10.96537	11.46238	12.57132	13.90396	15.53763	17.56477
1.3	10.52099	10.96135	11.45323	12.55095	13.87060	15.48698	17.49407
1.35	10.52155	10.95866	11.44669	12.53596	13.84599	15.44927	17.44093
1.4	10.52229	10.95681	11.44179	12.52428	13.82658	15.41923	17.39862
1.45	10.52320	10.95581	11.43857	12.51606	13.81238	15.39702	17.36708
1.5	10.52423	10.95560	11.43684	12.51091	13.80342	15.38253	17.34551
1.55	10.52544	10.95600	11.43631	12.50817	13.79797	15.37327	17.33154
1.6	10.52679	10.95700	11.43695	12.50782	13.79625	15.36952	17.32507
1.65	10.52831	10.95859	11.43864	12.50963	13.79815	15.37105	17.32556
1.7	10.52996	10.96066	11.44129	12.51338	13.80282	15.37671	17.33183
1.75	10.53171	10.96296	11.44434	12.51787	13.80862	15.38397	17.34022
1.8	10.53365	10.96589	11.44845	12.52441	13.81777	15.39608	17.35497
1.85	10.53565	10.96893	11.45282	12.53145	13.82754	15.40910	17.37104
1.9	10.53778	10.97231	11.45776	12.53957	13.83882	15.42430	17.39003
1.95	10.54011	10.97604	11.46323	12.54860	13.85163	15.44160	17.41172
2	10.54249	10.98013	11.46940	12.55908	13.86670	15.46222	17.43783

**Table 5 nanomaterials-12-03577-t005:** The effective electron density of the investigated samples at the studied gamma energy range.

Energy (MeV)	N_eff_ (×10^23^ Electrons g^−1^)						
	**PC-Bi_2_O_3_ wt %**						
	**0%**	**5%**	**10%**	**20%**	**30%**	**40%**	**50%**
0.1	3.18135	5.19805	6.09298	6.91258	7.18955	7.18831	6.99288
0.15	3.04590	4.48164	5.45172	6.51085	6.95042	7.05541	6.93421
0.2	2.99822	3.87816	4.64335	5.76028	6.37867	6.64266	6.64753
0.25	2.97752	3.53156	4.06047	4.97826	5.64974	6.04755	6.19652
0.3	2.96726	3.34209	3.70713	4.39197	4.97886	5.41354	5.66645
0.35	2.96137	3.22693	3.49181	3.99969	4.46780	4.86325	5.14907
0.4	2.95772	3.15657	3.35032	3.73745	4.10196	4.43483	4.70453
0.45	2.95558	3.10915	3.25975	3.55709	3.84640	4.11791	4.35354
0.5	2.95417	3.07568	3.19481	3.43197	3.66260	3.88211	4.08472
0.55	2.95254	3.05051	3.14625	3.33883	3.52890	3.70979	3.88020
0.6	2.95174	3.03122	3.10831	3.26742	3.42298	3.57265	3.71445
0.65	2.95085	3.01582	3.07994	3.21149	3.34036	3.46325	3.58346
0.7	2.95020	3.00407	3.05854	3.16765	3.27508	3.38048	3.47956
0.75	2.94966	2.99439	3.04081	3.13115	3.22148	3.31028	3.39490
0.8	2.94926	2.98664	3.02650	3.10171	3.18009	3.25504	3.32621
0.85	2.94894	2.98016	3.01451	3.07717	3.14487	3.20843	3.26773
0.9	2.94868	2.97475	3.00450	3.05791	3.11524	3.16910	3.22025
0.95	2.94845	2.97001	2.99567	3.04094	3.08907	3.13415	3.17795
1	2.94827	2.96610	2.98843	3.02689	3.06724	3.10510	3.14259
1.05	2.94810	2.96276	2.98205	3.01455	3.04829	3.08102	3.11154
1.1	2.94797	2.95994	2.97635	3.00348	3.03103	3.05930	3.08445
1.15	2.94788	2.95770	2.97174	2.99448	3.01718	3.04167	3.06277
1.2	2.94785	2.95577	2.96776	2.98664	3.00565	3.02608	3.04357
1.25	2.94788	2.95431	2.96466	2.98047	2.99667	3.01381	3.02836
1.3	2.94799	2.95322	2.96229	2.97564	2.98948	3.00399	3.01617
1.35	2.94814	2.95250	2.96060	2.97209	2.98418	2.99667	3.00700
1.4	2.94835	2.95200	2.95933	2.96931	2.98000	2.99085	2.99971
1.45	2.94861	2.95173	2.95850	2.96737	2.97694	2.98654	2.99427
1.5	2.94890	2.95167	2.95806	2.96615	2.97501	2.98373	2.99055
1.55	2.94923	2.95178	2.95792	2.96550	2.97383	2.98193	2.98814
1.6	2.94961	2.95205	2.95808	2.96541	2.97346	2.98121	2.98703
1.65	2.95004	2.95248	2.95852	2.96584	2.97387	2.98150	2.98711
1.7	2.95050	2.95304	2.95921	2.96673	2.97488	2.98260	2.98819
1.75	2.95099	2.95366	2.95999	2.96780	2.97613	2.98401	2.98964
1.8	2.95154	2.95445	2.96106	2.96935	2.97810	2.98636	2.99218
1.85	2.95209	2.95527	2.96219	2.97101	2.98021	2.98888	2.99496
1.9	2.95269	2.95618	2.96347	2.97294	2.98264	2.99183	2.99823
1.95	2.95335	2.95718	2.96488	2.97508	2.98540	2.99519	3.00197
2	2.95401	2.95828	2.96648	2.97757	2.98864	2.99919	3.00647

**Table 6 nanomaterials-12-03577-t006:** Fast neutron effective removal cross sections of the samples using Geant4 compared to manually calculated ones.

Element	Weight Fraction	Σ*_R_*/*ρ* (cm^2^/g)	Partial Density (g/cm^3^)	Σ*_R_* (cm^−1^) Calculation	Σ*_R_* (cm^−1^) Genat4	Diff%
**PC-Bi_2_O_3_ (0%)**						
**C (Carbon)**	0.279420	0.050187	0.321333	0.016127	**0.035317**	**8.40%**
**P (Phosphorus)**	0.720580	0.027066	0.828667	0.022429		
				**0.038556**		
**PC-Bi_2_O_3_ (5%)**						
**C (Carbon)**	0.265449	0.050187	0.315884	0.015853	**0.035329**	**8.71%**
**O (Oxygen)**	0.005150	0.040528	0.006129	0.000248		
**P (Phosphorus)**	0.684551	0.027066	0.814616	0.022048		
**Bi (Bismuth)**	0.044850	0.010295	0.053371	0.000549		
				**0.038699**		
**PC-Bi_2_O_3_ (10%)**						
**C (Carbon)**	0.251478	0.050187	0.311833	0.015650	**0.036066**	**7.71%**
**O (Oxygen)**	0.010301	0.040528	0.012773	0.000518		
**P (Phosphorus)**	0.648522	0.027066	0.804167	0.021765		
**Bi (Bismuth)**	0.089699	0.010295	0.111227	0.001145		
				**0.039078**		
**PC-Bi_2_O_3_ (20%)**						
**C (Carbon)**	0.223536	0.050187	0.304009	0.015257	**0.038126**	**4.98%**
**O (Oxygen)**	0.020602	0.040528	0.028019	0.001136		
**P (Phosphorus)**	0.576464	0.027066	0.783991	0.021219		
**Bi (Bismuth)**	0.179398	0.010295	0.243981	0.002512		
				**0.040124**		
**PC-Bi_2_O_3_ (30%)**						
**C (Carbon)**	0.195594	0.050187	0.301215	0.015117	**0.040868**	**3.74%**
**O (Oxygen)**	0.030903	0.040528	0.04759	0.001929		
**P (Phosphorus)**	0.504406	0.027066	0.776785	0.021024		
**Bi (Bismuth)**	0.269097	0.010295	0.414409	0.004266		
				**0.042336**		
**PC-Bi_2_O_3_ (40%)**						
**C (Carbon)**	0.167652	0.050187	0.291714	0.014640	**0.042375**	**4.42%**
**O (Oxygen)**	0.041204	0.040528	0.071695	0.002906		
**P (Phosphorus)**	0.432348	0.027066	0.752286	0.020361		
**Bi (Bismuth)**	0.358796	0.010295	0.624305	0.006427		
				**0.044334**		
**PC-Bi_2_O_3_ (50%)**						
**C (Carbon)**	0.139710	0.050187	0.275229	0.013813	**0.046089**	**0.31%**
**O (Oxygen)**	0.051505	0.040528	0.101464	0.004112		
**P (Phosphorus)**	0.360290	0.027066	0.709771	0.019210		
**Bi (Bismuth)**	0.448495	0.010295	0.883535	0.009096		
				**0.046231**		

**Table 7 nanomaterials-12-03577-t007:** Thicknesses of the samples equivalent to 0.5 mm of lead.

PC wt %	Bi_2_O_3_ wt %	0.5 mm-Lead-Equivalent Thickness (cm)
100	0	0.907391
95	5	0.843263
90	10	0.779371
80	20	0.661724
70	30	0.546770
60	40	0.454661
50	50	0.378681

**Table 8 nanomaterials-12-03577-t008:** Energy deposits in the head, trunk, and whole body of the phantom.

	Air	Pb	PC-Bi_2_O_3_ wt %						
			0%	5%	10%	20%	30%	40%	50%
**Head**	6.32000	7.46590	7.33951	7.53072	7.61248	9.55394	7.49214	7.25332	8.18214
**Trunk**	1119.68	1109.81	1051.73	1046.49	1058.09	1030.42	1030.02	1035.96	1028.44
**BODY**	1432.98	1428.78	1341.16	1343.52	1352.23	1334.79	1319.8	1330.9	1330.86

## Data Availability

Not applicable.

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
