# Peer review of "Theoretical Investigation of Fast Neutron and Gamma Radiation Properties of Polycarbonate-Bismuth Oxide Composites Using Geant4"

_nanomaterials, 2022, doi:10.3390/nano12203577_

Round 1

Reviewer 1 Report

Dear Editor

The manuscript titled "Theoretical Investigation of Fast Neutron and Gamma Radia- 2 tion Properties of Polycarbonate-Bismuth Oxide composites us- 3 ing Geant4" Is very interesting. Personally, I found it interesting research that I enjoyed reading. The article describes in a clear and orderly way the tests carried out and describes the results obtained.

Not being an expert in the field of polymers and radiation, I still consider the work well written.

In the work the nano dimension of the Bismuth compound used is mentioned several times, without however being given its characterization. Is it Nanometric? Is it crystalline, or is it amorphous? what shape, what size does it have? I believe that an upgrade (for example TEM Image, XRD Diffraction...) in this sense is useful to better characterize and understand what material we are talking about and which gives the properties to the polymer studied.

Reviewer 3 Report

This is a paper investigating a model for nuclear physics using Geant4. The results of using and applying the program to derive data are judged to be meaningful.

It can inform readers of the application of the program and is considered to be a helpful resource for nano research related to nuclear physics.
